# Port-a-Cath Infection of *Mycobacterium senegalense*: First Italian Case Report

**DOI:** 10.3390/microorganisms12122431

**Published:** 2024-11-26

**Authors:** Giulia Grassia, Francesco Amisano, Stefano Gaiarsa, Jessica Bagnarino, Francesca Compagno, Arianna Panigari, Fausto Baldanti, Vincenzina Monzillo, Daniela Barbarini

**Affiliations:** 1Department of Microbiology & Virology, Fondazione IRCCS Policlinico San Matteo, 27100 Pavia, Italy; g.grassia@smatteo.pv.it (G.G.); s.gaiarsa@smatteo.pv.it (S.G.); j.bagnarino@smatteo.pv.it (J.B.); f.baldanti@smatteo.pv.it (F.B.); vincenzinamonzillo@gmail.com (V.M.); d.barbarini@smatteo.pv.it (D.B.); 2Specialization School of Microbiology and Virology, University of Pavia, 27100 Pavia, Italy; 3Pediatric Hematology/Oncology, Fondazione IRCCS Policlinico San Matteo, 27100 Pavia, Italy; f.compagno@smatteo.pv.it (F.C.); a.panigari@smatteo.pv.it (A.P.); 4Department of Clinical, Surgical, Diagnostic and Pediatric Sciences, University of Pavia, 27100 Pavia, Italy

**Keywords:** catheter-associated infection, non-tuberculous mycobacteria, NTM, Port-a-Cath infection

## Abstract

*Mycobacterium senegalense* is a Non-Tuberculous Mycobacterium (NTM) belonging to the *M. fortuitum* group, often associated with veterinary diseases, such as bovine farcy. However, it can also cause human infections and appears to be involved in Catheter-Associated Infections in immunocompromised patients. Here, we report the first Italian isolation of a strain of *M. senegalense* from a 16-year-old oncological female patient being treated at Fondazione IRCCS Policlinico San Matteo Pavia (Italy). Following pain at the Port-a-Cath site, a pus culture was collected and the positivity for the *M. fortuitum* group revealed the NTM infection. Antimicrobial susceptibility tests were performed and interpreted according to the available CLSI breakpoints. This information allowed us to implement the correct antibiotic therapy that, together with the device removal, led to the patient’s recovery. Finally, due to the increasing number of isolations, the possible presence of NTM infections in prosthetic devices should be among the primary diagnostic questions in a clinical setting.

## 1. Introduction

*Mycobacterium senegalense* is a non-tuberculous rapidly growing mycobacterium (RGM) that belongs to the *Mycobacterium fortuitum* group, a monophyletic group including several pathogens, such as *Mycobacterium fortuitum sensu stricto*, *Mycobacterium farcinogenes*, *Mycobacterium peregrinum* and *Mycobacterium porcinum*, all responsible for a wide set of infections [1]. *M. senegalense* is known as the main pathogen of bovine farcy, which manifests as a chronic suppurative granuloma of the skin and superficial lymphatics, particularly in cattle in East and Central Africa. *M. senegalense* was originally described by Chamoiseau in 1973 as a subspecies of *M. farcinogenes* [2]. However, it was later recognised as a distinct species closely related to *M. fortuitum* [3].

Until now, few human cases have been described, more frequently in immunocompromised patients [4,5,6]. This opportunistic pathogen causes post-traumatic skin and soft tissue infections, including postsurgical wound infections.

*M. senegalense* as well as other RGM have also been reported to be responsible for infections of prosthetic devices, such as catheters, joint prostheses, breast implants and others [7,8,9].

Non-Tuberculous Mycobacteria (NTM) are generally known to be acquired from the environment through ingestion, inhalation and skin contact, and may cause lymphadenitis, pulmonary and disseminated infections as well as skin and soft tissue infections [10].

Several studies confirm that these microorganisms have the ability to adhere to biomaterials by forming biofilms. This pathogenic factor makes them particularly resistant to the immune system and antibiotics; once colonised, the device must necessarily be removed, as antibiotic therapy alone cannot eliminate the infection [11,12,13].

The first case of human infection sustained by *M. senegalense* was described in 2005 in Korea for causing a catheter-related bloodstream infection in a cancer patient [4].

Establishing an accurate diagnosis of *M. senegalense* infection is incredibly difficult, because its symptoms are non-specific, requiring histological examination and extensive mycobacterial cultures [6,14].

Routine diagnostic tests are not very sensitive to give correct sub-specie identification and specific phenotypic and genotypic techniques, not within the reach of all laboratories, have to be performed [15]. Indeed, *M. senegalense* turns out to be genetically very similar to *M. farcinogenes*, so molecular methods must be supplemented with cultural and biochemical ones in order to identify it correctly [16].

Here, we describe the first case in Italy of a Port-a-Cath central catheter infection by *M. senegalense* in a young girl with oncological pathology admitted to Fondazione IRCCS Policlinico San Matteo Pavia (Italy).

## 2. Case Description

We were presented with a 16-year-old girl diagnosed in November 2021 with a right paravertebral, non-metastatic, adult-type epithelial sarcoma, a malignant non-rhabdomyosarcoma soft tissue sarcoma (NRSTS) that occurs mainly in adulthood and is characterised by an uncertain response to chemotherapy. Given the local aggressiveness and metastatic tendency of this tumour, our patient was treated with a multimodal therapeutic approach including surgery, chemotherapy and radiotherapy, with a very good partial response to treatment. However, in May 2023, eight months after completion of first-line therapy, she was diagnosed with local and metastatic disease recurrence. She therefore received reinduction salvage chemotherapy after implantation of a Port-a-Cath Central Venous Catheter (CVC) in the pectoral region, below the left clavicle. She was closely followed in our outpatient clinic, receiving sulfamethoxazole/trimethoprim as prophylaxis for *Pneumocystis jirovecii*.

After the sixth course of reinduction chemotherapy, seven months after the CVC placement, during the follow-up visit, the patient complained of a two-day history of pain at the CVC site; no fever or chills were reported. Physical examination yielded no significant findings except for minimal dehiscence of the 2 cm surgical wound at the insertion site of the reservoir of the Port-a-Cath, with purulent discharge (Figure 1). In addition, we observed erythema, localised edoema and soft tissue induration around the catheter reservoir site.

Blood cultures and exudate samples from the lesion were taken for microscopic examination and microbial cultures. The only laboratory findings of note were mild lymphopenia with a normal white blood cell count and normal neutrophils. C-reactive protein was slightly elevated at 11.7 mg/dL (normal range < 0.5 mg/dL). Empirical antimicrobial therapy was started in our outpatient department with intravenous ceftriaxone and teicoplanin. Three days later, as there was no improvement, treatment was changed to oral clindamycin and ciprofloxacin. After 7–10 days, exudate culture resulted positive for the *M. fortuitum* group, while blood cultures remained negative.

Once susceptibility results became available (Table 1), therapy was changed to oral doxycycline and a higher dose of ciprofloxacin. At the same time, the patient underwent surgical removal of the Port-a-Cath CVC and curettage of the involved skin area. Workup for disseminated infection included ultrasound evaluation of the cervical and axillary lymph nodes and mycobacterial blood cultures, which were negative. However, the culture of the skin around the insertion site of the reservoir of the CVC confirmed the presence of the *M. fortuitum* group, corroborating the diagnosis of a NTM device-related infection. Overall, the patient received a 1-month course of antimicrobial therapy for the *M. fortuitum* group after CVC removal, with complete resolution of the infection. A comprehensive investigation was conducted to identify potential sources of infection, yet no definitive conclusions could be drawn. The patient demonstrated acceptable standard of living and personal hygiene at home, and there was no reported contact with animals. Additionally, there was no history of recent foreign travel. The CVC was placed in the operating room, and the dressing was changed on a weekly basis.

## 3. Microbiological Analyses

Exudate samples from the lesion were grown on 5% sheep’s blood agar (bioMérieux^®^, Marcy-l’Étoile, France), in the BACTEC MGIT medium (Becton Dickinson and Company, Franklin Lakes, NJ, USA) and on Löwenstein-Jensen Medium (Thermo Fisher Scientific™, Waltham, MA, USA). After incubation at 37 °C for 48 h in 5% sheep’s blood agar, the growth of white and wrinkled colonies was observed from all clinical samples. With a longer time frame, mycobacteria-specific cultures also became positive. The grown colonies were identified as the *M. fortuitum* group by using Matrix-Assisted Laser Desorption Ionisation Time-Of-Flight (MALDI-TOF) mass spectrometry (Bruker Daltonics GmbH, Bremen, Germany). Identification was completed by performing molecular tests, such as the GenoType CM (Hain Lifesciences/Arnika, Nehren, Germany), which confirmed the *M. fortuitum* group identification and by sequencing the whole genome of the isolate with the Illumina MiSeq platform.

The genome 35FGF was assembled using Shovill 1.1 (https://github.com/tseemann/shovill, accessed on 13 May 2024) and it is available in the NCBI database under Bioproject PRJNA1164745. The correct identification at species level was made difficult by an inaccurate annotation of genomes available on NCBI. In fact, 35FGF presented a high homology with strains deposited as *M. farcinogenes*, *M. conceptionense*, and *M. senegalense*, thus making difficult the correct discrimination. In addition, the species annotation of genomes registered as *M. farcinogenes* was recently questioned, as the reference genome for this species (DSM 43637) was recognised as *M. senegalense* by Turenne and colleagues [17].

For this reason, we compared the sequences of the ITS region of our strain with database sequences [16]. We obtained the highest identity values with *M. senegalense* (92% identity, 100% query cover), *M. conceptionense* (86% identity, 100% query cover), and *M. farcinogenes* (88% identity, 63% query cover). We further investigated the species-level identification of our strain by performing a phylogenomic analysis on all *M. senegalense* and all *M. conceptionense* genomes available in the BV-BRC database last accessed on 24 September 2024 (https://www.bv-brc.org). In detail, we obtained a concatenation of ortholog genes using panaroo [18] and inferred the phylogeny using iqtree [19]. The resulting phylogenetic tree (Figure 2) indicated that isolate 35FGF was enclosed in a highly supported monophylum with *M. conceptionense* genomes. However, the genomes of the two species are not separated by the tree topology. Notably, *M. conceptionense* was recently suggested to be part of the *M. senegalense* species [20]. Our resulting tree topology is in agreement with the literature; we thus assigned our isolate to the *M. senegalense* species.

Furthermore, additional phenotypic tests, addressing morphological and cultural characteristics, were performed to better discriminate *M. senegalense* from *M. farcinogenes.* In particular, colonies appeared after 3 days of incubation at 37 °C on the Löwenstein Jensen medium and were nonchromogenic, wheat-coloured, and relatively emulsifiable, properties that are typical of *M. senegalense* [15].

At the same time, the antimicrobial profile of the clinical isolate was determined by Minimum Inhibitory Concentration (MIC), according to the Clinical and Laboratory Standards Institute (CLSI) [21].

To perform MICs, we used a Sensititre RAPMYCOI AST plate (Thermo Fisher Scientific™). *M. senegalense* clinical isolate showed susceptibility to ciprofloxacin (0.5 µg/mL), moxifloxacin (≤0.25 µg/mL), amikacin (≤1 µg/mL), doxycycline (0.5 µg/mL), clarithromycin (0.25 µg/mL), linezolid (4 µg/mL), imipenem (4 µg/mL), resistance to sulfamethoxazole/trimethoprim (8/152 µg/mL), and intermediate sensitivity to cefoxitin (32 µg/mL) and tobramycin (4 µg/mL) (Table 1). The quality control of the tests was performed with the *Staphylococcus aureus* ATCC^®^ 29213 strain, as recommended by the CLSI document [21].

## 4. Discussion and Conclusions

In this case report, we document a wound and Port-a-Cath infection caused by *M. senegalense* in a 16-year-old girl with a malignant NRSTS. *M. senegalense* is a RGM belonging to the *M. fortuitum* group, and like other RGM such as the *M. smegmatis* group, and the *M. chelonae/abscessus* group, is capable of thriving in even the most hostile environments [11]. Due to their ubiquitous distribution, RGM have been identified worldwide with increasing frequency causing human infections. Verifying sources of infection requires the identification of an identical genotype between clinical and environmental isolates. Identifying transmission routes and sources of infection is particularly difficult because the diseases they cause often occur after long incubation periods [10]. However, RGM have been recognised as one of the significant pathogens of prosthetic devices infections, due to their capability to colonise and form biofilm on abiotic surfaces [22]. Notably, among RGM, the *M. fortuitum* group has been the most frequently mycobacterial pathogen associated with catheter infections in immunocompromised patients [4,11,12].

The *M. fortuitum* group includes several species, such as *M. fortuitum* sensu stricto, *M. peregrinum*, *M. mucogenicum*, *M. mageritense*, *M. farcinogenes*, and *M. senegalense* (and several newly described species such as *M. septicum*, *M. houstonense*, *M. boenickei*, *M. neworleansense*, and *M. brisbanense*) [4]. Among these, *M. senegalense* was originally described by Chamoiseau in 1973 as a subspecies of *M. farcinogenes*. Although *M. farcinogenes* and *M. senegalense* have identical 16S rRNA gene sequences, *M. senegalense* could be identified as a different species based on differences in growth rate, chemical activity and a distinction based on the ITS region has been proposed [1,15,16]. However, the lack of availability of referenced *M. farcinogenes* genomes does not allow us to correctly discriminate between these two species.

While most species of the *M. fortuitum* group have been reported to be responsible for various human diseases, human infections determined by *M. senegalense* have been described only in a few cases [4,6]. This is most likely due to the placement of *M. senegalense* in the *M. fortuitum* group without a specific identification at species-level, or eventually to a misidentification as *M. farcinogenes*, like in the case of DSM 43637 [17,23].

*M. senegalense*, together with *M. farcinogenes*, is recognised as the main etiological agent of bovine farcy, a disease that causes a chronic suppurative granulomatous inflammation of the skin and lymphatic system in cattle [12]. Although the human pathogenicity of this NTM species has not been fully elucidated, reports including *M. senegalense* strains responsible for integumentary, hematologic, or musculoskeletal system infections have recently appeared [4,5,15]. These cases involved both immunocompromised and immunocompetent patients without any predominance in particular age brackets. The accurate identification, together with the availability of the relative drug-susceptibility profile, is undoubtedly essential for clinicians to begin the correct therapy and thus to increase the probability of a successful outcome [8].

In our clinical case, a few days after the patient reported pain at the Port-a-Cath site, empirical therapy with intravenous ceftriaxone and teicoplanin was started. After three days without improvement, it was decided to replace the therapy with oral clindamycin and ciprofloxacin. In the meantime, we isolated and identified a *M. fortuitum* group from pus culture, sensitive to the drugs shown in Table 1. After reviewing the antibiogram, it was decided to optimise therapy by replacing clindamycin with oral doxycycline. At the same time, the Port-a-Cath device, most likely colonised by the *M. fortuitum* group via biofilm production, was removed. After one month of therapy with the two drugs in combination, complete resolution of the infection was achieved.

Meanwhile, the genome 35FGF responsible for the infection was sequenced and identified as *M. senegalense* thanks to a phylogenomic analysis.

The increase in NTM diseases can be explained by several contributing factors, such as improvements in laboratory detection techniques, changes in diagnostic criteria, and raised awareness of NTM diseases in the clinical field [24].

We can therefore conclude that in prosthetic device infections, it is also useful to investigate NTM as a possible aetiological agent, particularly in immunocompromised patients. For this reason, an accurate and timely identification, as well as the investigation of the antimicrobial susceptibility profile of fast-growing nontuberculous mycobacteria species, are crucial for establishing the prompt and correct treatment of the infection.

## Figures and Tables

**Figure 1 microorganisms-12-02431-f001:**
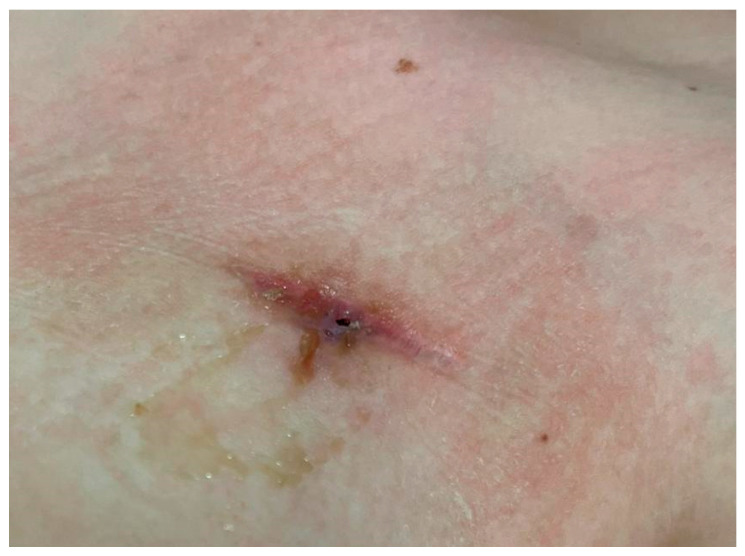
Wound at the Port-a-Cath Central Venous Catheter site.

**Figure 2 microorganisms-12-02431-f002:**
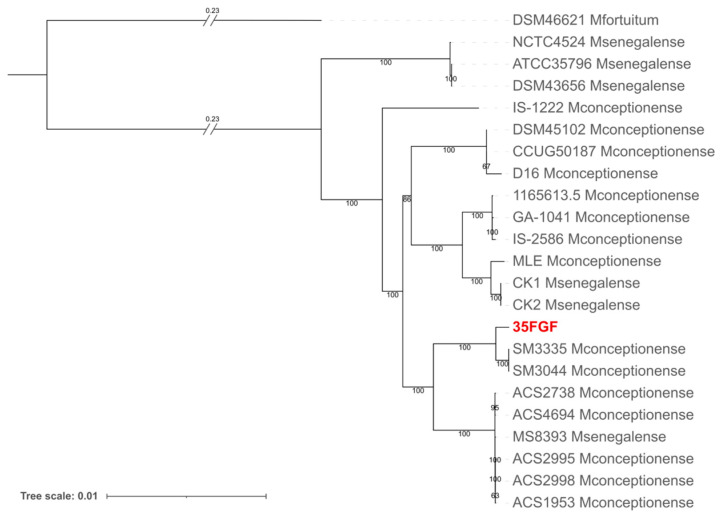
Phylogenetic analysis obtained with iqtree of the 35FGF isolate and all 21 high-quality *M. senegalense* genomes (including those submitted as *M. conceptionense*) available in the BV-BRC database (last accessed on 24 September 2024). *M. fortuitum* was also included as an outgroup.

**Table 1 microorganisms-12-02431-t001:** Pattern of antibiotic susceptibility of *M. senegalense*.

Antibiotic	MIC (µg/mL)	Interpretation *
Amikacin	≤1	S
Amoxicillin clavulanate	>64/32	NI
Cefepime	>32	NI
Cefoxitin	32	I
Ceftriaxone	>64	NI
Ciprofloxacin	0.5	S
Clarithromycin	0.25	S
Doxycycline	0.5	S
Imipenem	4	S
Linezolid	4	S
Minocycline	≤1	NI
Moxifloxacin	≤0.25	S
Tigecycline	0.25	NI
Tobramycin	4	I
Trimethoprim/sulfamethoxazole	8/152	R

* Interpretation was performed according to the Clinical and Laboratory Standards Institute (CLSI) breakpoints. S: Susceptible; I: Intermediate; R: Resistant; NI: no CLSI guidelines for this antibiotic/organism combination.

## Data Availability

The genome is available from the NCBI database under Bioproject PRJNA1164745.

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
