# Peer review of "Port-a-Cath Infection of Mycobacterium senegalense: First Italian Case Report"

_microorganisms, 2024, doi:10.3390/microorganisms12122431_

Round 1

Reviewer 1 Report

Comments and Suggestions for Authors

The article is original and within the Journal aim. Many thanks as the manuscript is clear and focused. I would like to make some minor and constructive observations about your work.

line 32 please revise the typo with farcy

line 72-76 Please specify if the purulent discharge is from an unhealed surgical incision or from a subsequent needle puncture. Moreover, also specify the time elapsed from the implantation of the device to the onset of symptoms. In addition, any risky domestic activities, recent travel and contact with risky animals may merit investigation, of course if these data are available as absolutely atypical case and presentation.

line 164 and 201. Please hypothesise in relation also to the background from line 42 a possible route of infection, as devices must be sterile, with the subsequent need to investigate the route of contamination. Verify if in the literature mentioned this topic was deepened. I propose to cite doi.org/10.3390/healthcare12171788 which set the state of the art on the issue of NTM infections and their relevance and importance in healthcare, but also to advocate for research on the issue of modality of transmission to humans.

Informed Consent Statement: Informed consent was obtained from the patient involved in the
study. Please specify relatives' consent and minor agreement as a 16 years old subject.
many thanks

Reviewer 2 Report

Comments and Suggestions for Authors

In the following article entitled "Port-a-Cath infection of Mycobacterium senegalense: first Italian case report", the authors report the first Italian isolation of a strain of Mycobacterium senegalense from a 16-year-old oncological female patient being treated at Fondazione IRCCS Policlinico San Matteo Pavia. Antimicrobial susceptibility tests were performed and interpreted according to the available CLSI breakpoints. This information allowed to implement the correct antibiotic therapy that, together with the device removal, led to the patient’s recovery.

After careful reading of the manuscript, I recommend publication in microorganisms after the following very minor revisions have been performed.

The investigation performed is indeed of great interest and quite original, and the manuscript well written.

For a better soundness, it deserves only adding few extra references, such as, for instance:

Amin-Nordin, S. et al., Mycobacterium senegalense catheter-related bloodstream infection, at Infectious diseases.

Dos Santos, L. S.; et al. Prosthetic joint infection caused by an imipenem-resistant Mycobacterium senegalense, Clinical Microbiology, 2023, 54, 929‒934.

Please, also modify L21: ….,the possible presence of NTM…..
